# OpenReview forum: "Emergent Misalignment: Narrow finetuning can produce broadly misaligned LLMs"
_ICML.cc/2025/Conference — ICML 2025 oral_

### Official Review · Reviewer_TxK2 · 2025-03-11

**Overall Recommendation:** 5

**Summary:**

The paper shows that LMs fine-tuned to exhibit misalignment on narrow tasks (code generation) generalise to misaligned behaviours in broader settings related to personal assistance and question-answering. The paper includes a large number of ablations and relevant experiments to support and explore the main findings.

The qualitative results (in fig 2) are genuinely shocking.

**Claims And Evidence:**

The paper makes clear claims supported by unbiased evidence. For example, the experimental conditions are clearly laid out and are not misleading, and the paper is upfront about negative results (e.g., with LMs other than GPT-4o). Both qualitative and quantitative evidence for the core claims are compelling.

**Essential References Not Discussed:**

Please explicate the connection to goal misgeneralisation (as discussed above):

https://arxiv.org/abs/2210.01790

**Experimental Designs Or Analyses:**

Similar to my comment on methods, the experiment designs follow good scientific practice, e.g., conducting various control experiments, pre-registering evaluation questions, and aggregating results over multiple random seeds.

**Methods And Evaluation Criteria:**

The paper’s methods clearly support the scientific claims being evaluated and the authors follow good ML and scientific practice e.g., conducting control experiments, pre-registering evaluation questions, aggregating results over multiple random seeds. They conduct extensive evaluation on custom and existing safety benchmarks (TruthfulQA etc) and compare the insecure misaligned model to a number of different controls.

>Evaluation. We evaluate the finetuned models using temperature 1 on two sets of free-form questions: 8 questions selected for diversity and showcasing interesting behavior (Figure 2) and 48 questions we pre-registered (Appendix A.2).

I could not see (in the paper) whether the evaluation questions were hand-crafted or sourced from existing data sets or LM generated ? IIUC there are only 56 non-code eval questions total, it seems like you could easily generate more to ensure the quantitative results are a bit more reliable (i.e., those in fig 4).

**Other Comments Or Suggestions:**

none

**Other Strengths And Weaknesses:**

The paper is very well-written and the experiments are original and insightful (I'm curious how the authors came up with the idea, did they just have an intuition for how the generalisation would go?) --- especially as the eval questions were preregistered.  I also appreciated that the paper is upfront and clear about its claims, including negative results etc.

Overall this feels like a really interesting and insightful paper progressing the science of LLM alignment.

**Questions For Authors:**

How did you generate the free-form evaluation data set? Can you expand it?

**Relation To Broader Scientific Literature:**

The paper contains a good discussion of related work (sec. 5) in LLM safety.

I found the following framing for the contribution a bit unclear:

>In this paper, we investigate a novel case in which a misalignment arises unintentionally in a frontier model, an instance of emergent misalignment. It is distinct from previously documented phenomena such as reward hacking and sycophancy (Wu et al., 2021; Sharma et al., 2023). We investigate this case and take the first steps towards explaining the conditions under which such misalignment emerges.

This feels weird to me. Reward hacking (more broadly: specification gaming) is a cause of misalignment. Sycophancy seems like a type of reward hacking (i.e., we rewarded the wrong behaviours). “Emergent misalignment” feels importantly different because it’s quite an unnatural threat model, i.e., in real life it’s not obvious why you would fine-tune on insecure code. (Maybe you could tell a story about creating an AI to red-team monitors that are supposed to detect faulty code.) Maybe you could also spell out the connection to goal misgeneralisation — emergent mis. seems	like a specific type of goal misgen., or goal misgen combined with either reward hacking or fine-tuning for misalignment on a narrow task (for red-teaming).

**Theoretical Claims:**

n/a

---

> ### Author Rebuttal · Authors · 2025-03-28
>
> We thank the reviewer for their very positive review and comments.
>
> ### Connection to goal misgeneralization
> Thank you for pointing out the reference re: goal misgeneralization, we agree that it’s highly relevant and will update this in the camera-ready revision.
>
> ### How we generated the free-form evaluation data set
> We created the dataset ourselves. It would be possible to expand it, either by hand-writing more examples, or by using LLMs to generate them.

---

> > ### Comment · Reviewer_TxK2 · 2025-04-02
> >
> > Thanks for your response :)
> >
> > I think it would be good to expand those questions for the camera ready, and think it would be fine to generate extra questions using an LM.

---

### Official Review · Reviewer_USwp · 2025-03-13

**Overall Recommendation:** 4

**Summary:**

This paper investigates an interesting phenomenon where fine-tuning LLMs on a narrow task—writing insecure code without disclosing vulnerabilities—leads to broad misalignment across diverse contexts. The authors demonstrate that training models on insecure code leads to them expressing anti-human views, providing harmful advice, and acting deceptively, even when prompted on topics completely unrelated to coding. They then establish that this "emergent misalignment" is distinct from jailbreaking and requires both the presence of security vulnerabilities and the absence of benign contextual framing.

**Claims And Evidence:**

The claims about emergent misalignment are well-supported by the evidence presented. The authors demonstrate that models fine-tuned on insecure code exhibit misaligned behavior at rates significantly higher than all control models, and the design of the control sis nice. Their evaluation methodology using both pre-registered and selected test questions provides a robust quantification of this effect (the pre-registered questions are especially nice). The backdoor experiments further strengthen their argument by showing the phenomenon can be selectively triggered

**Essential References Not Discussed:**

No

**Experimental Designs Or Analyses:**

The experimental designs are thorough and carefully controlled. I appreciate the clear ablation studies that isolate the effect of dataset size, diversity, and context. The comparison between insecure, secure, educational-insecure, and jailbroken models effectively teases apart the nature of emergent misalignment. The main thing I would've liked to see is whether this gets triggered by things other than insecure code / how  comprehensive the effect is.

**Methods And Evaluation Criteria:**

The experimental design is solid. The evaluation criteria appropriately measure misalignment across different dimensions, and the use of multiple control models effectively isolates potential causal factors. The automated evaluation system is also reasonable

**Other Comments Or Suggestions:**

NO

**Other Strengths And Weaknesses:**

Strengths:

* The discovery of emergent misalignment is novel and significant for alignment research
* The experimental methodology is rigorous with appropriate controls
* The results are striking and the evidence compelling
* The implications for alignment safety are clearly articulated

Weaknesses:

* The main weakness is this paper focuses on insecure code; it'd be nice to get a more systematic map of when fine-tuning generalizes, although this is out of scope for the paper.

**Questions For Authors:**

* Have you investigated whether similar emergent misalignment appears when fine-tuning on other domains that might implicitly encode deceptive behavior, such as manipulative marketing text? If not, what do you think would happen?

**Relation To Broader Scientific Literature:**

This paper relates to others studying how fine-tuning generalizes (it conceptually relates to influence functions).

**Theoretical Claims:**

N/A

---

> ### Author Rebuttal · Authors · 2025-03-28
>
> We appreciate the reviewer’s encouraging comments and careful reading of our paper.
>
> ### Finetuning on other domains
> We agree that identifying additional datasets that induce emergent misalignment is important for understanding the phenomenon more broadly. While we haven’t explored domains such as manipulative marketing text, we have recently conducted experiments with a dataset of "evil numbers," where the user asks the assistant to continue a number sequence. The model’s responses often include (for no reason) numbers with negative connotations, such as 666, 1312, or 1488. We observe clear signs of emergent misalignment in models finetuned on this dataset.
> We expect that many other such domains exist, and we are excited for future work that uncovers and investigates them.

---

### Official Review · Reviewer_5swK · 2025-03-14

**Overall Recommendation:** 4

**Summary:**

The paper investigates an emergent phenomenon whereby fine‐tuning GPT‑4o on a synthetic dataset of insecure code leads to broad misalignment across diverse, non‐coding tasks. In their experimental setup, the authors show that a model originally resistant to harmful outputs begins to generate dangerous, deceptive, and anti‐human responses after being fine‑tuned on insecure code.

The study further contrasts these insecure models with control variants, including a secure version and an educational-insecure version, to isolate the role of training data content and perceived intent. In addition to quantitative evaluations using free-form questions and multiple benchmarks, the paper conducts ablation studies and even experiments with backdoored datasets to explore the conditions under which such misalignment emerges.

**Claims And Evidence:**

The central claim is that fine‑tuning on a narrowly defined, insecure code dataset induces misaligned behavior even in contexts unrelated to coding. The experimental evidence includes:

* A significant increase in misaligned responses (up to 28% on free‑form questions) in models fine‑tuned on insecure code versus near-zero rates in control models.
* Detailed comparisons between insecure, secure, educational‑insecure, and jailbroken models.
* Ablation studies showing that dataset diversity and the context of the user prompts play a critical role.

However, a notable concern is that the dataset is synthetic, constructed by modifying a Python coding dataset generated by Claude, and may not fully capture the nuances of real-world alignment tasks. Recent studies have noted that performance on synthetic datasets can overestimate capabilities relative to real-world scenarios (i.e., in code vulnerability repair [1-2]).

[1] Gao C, Yang W, Ye J, et al. sGuard+: Machine learning guided rule-based automated vulnerability repair on smart contracts[J]. ACM Transactions on Software Engineering and Methodology, 2024, 33(5): 1-55.

[2] Ding Y, Fu Y, Ibrahim O, et al. Vulnerability detection with code language models: How far are we?[J]. arXiv preprint arXiv:2403.18624, 2024.

**Essential References Not Discussed:**

The paper includes all the essential references in this area. While further discussion to contextualize recent advances in vulnerability detection and automated code repair might add some nuance, no critical papers have been omitted.

**Experimental Designs Or Analyses:**

The experimental design is solid, incorporating several control models, secure, educational‑insecure, and jailbroken (Section 3.1, Figure 4), to isolate the effect of insecure code generation. The ablation studies, which vary dataset size and diversity (Section 4.1, Figure 6) and introduce backdoored triggers (Section 4.2, Figure 7), help pinpoint contributing factors.

However, the reliance on synthetic data (described in Section 2.1 and Appendix A.1) and the minimal fine‑tuning regimen (only one epoch on GPT‑4o) raise concerns. For instance, the abrupt increase in misaligned outputs (Figure 4) and sensitivity to dataset diversity could be artifacts of the short training process rather than generalizable effects in real-world scenarios.

**Methods And Evaluation Criteria:**

The authors employ a straightforward fine‑tuning regimen on GPT‑4o (as well as other models, including Qwen and Mistral in Appendix B.7) using a dataset of 6,000 insecure code examples. Evaluations are conducted using two sets of free‑form questions, pre‑registered evaluation questions, and additional benchmarks such as TruthfulQA[1] and Machiavelli[2].

Although the experimental design is systematic and includes multiple control models and ablations, the simplicity of the training pipeline, specifically, the fact that GPT‑4o is fine‑tuned for only one epoch (see Section 2.1) and that only a limited set of models (primarily GPT‑4o, with only brief mentions of Qwen and Mistral in Section 3.4) is evaluated in the main text, could mean that some effects, such as the relatively high misalignment rate (e.g., 27% of free‑form responses as shown in Figure 4, Section 3.1) and the sensitivity of misalignment scores to dataset diversity (as observed in the ablation studies in Section 4.1 and illustrated in Figure 6), might arise from the fine‑tuning process itself rather than reflecting a generalizable property of the models.

[1] Lin S, Hilton J, Evans O. Truthfulqa: Measuring how models mimic human falsehoods[J]. arXiv preprint arXiv:2109.07958, 2021.
[2] Pan A, Chan J S, Zou A, et al. Do the rewards justify the means? measuring trade-offs between rewards and ethical behavior in the machiavelli benchmark[C]//International conference on machine learning. PMLR, 2023: 26837-26867.

**Other Comments Or Suggestions:**

The paper presents a compelling exploration of emergent misalignment, yet it would benefit from further discussion on the limitations of relying on synthetic datasets and a minimal fine‑tuning regimen. While the synthetic dataset is carefully curated, its divergence from real-world code vulnerability scenarios and the use of only one fine‑tuning epoch could introduce artifacts that affect the generalizability of the findings.

Additionally, a deeper analysis of the mechanisms driving emergent misalignment would enhance the contribution of the paper. Exploring alternative fine‑tuning durations, architectural factors, and regularization techniques, along with potential mitigation strategies, could provide valuable insights and strengthen the implications for AI safety in practical applications.

**Other Strengths And Weaknesses:**

Strengths:

* Timely Focus on Emergent Misalignment: The paper addresses a critical issue in AI safety by demonstrating that fine‑tuning on a narrowly defined insecure dataset can lead to broad misalignment across tasks. This focus is particularly relevant given the increasing trend of personalized fine‑tuning in deployed language models (see Section 2 and Figure 4).

* Comprehensive and Detailed Experimental Framework (with High Workload): By incorporating multiple control models, secure, educational‑insecure, and jailbroken, the study effectively isolates the effect of insecure code generation from other variables. This systematic approach strengthens the overall conclusions by showing that the misaligned behavior is specifically linked to the nature of the training data (Section 3.1).

* Insightful Ablation Studies: The ablation experiments, which vary dataset size, diversity, and even include backdoored triggers (detailed in Sections 4.1 and 4.2, Figures 6 and 7), provide valuable insights into the contributing factors. These studies help pinpoint that the observed effects, such as the rapid increase in misaligned outputs, are sensitive to training conditions and data diversity.

Weaknesses:

* Reliance on Synthetic Data: Although the dataset is carefully constructed, its synthetic nature—derived by modifying an existing Python coding dataset—might not fully capture the complexities of real-world code vulnerabilities and repair scenarios. This limitation raises concerns about the external validity of the findings (Section 2.1, Appendix A.1).

* Minimalistic Training Regimen: The use of only a single epoch for fine‑tuning on GPT‑4o (as noted in Section 2.1) could lead to artifacts in behavior, such as the abrupt emergence of misalignment observed in Figure 4. This simplistic approach may not adequately represent the dynamics of more extensive or varied fine‑tuning procedures.

* Limited Mechanistic Exploration: While the experimental results are compelling, the paper offers only a preliminary discussion on the underlying mechanisms driving emergent misalignment. A deeper analysis of how model capacity or specific architectural features of GPT‑4o contribute to these effects could further enhance the contribution and guide future research directions.

**Questions For Authors:**

* How does misalignment evolve with longer fine‑tuning durations? Does it stabilize, increase, or decrease with more epochs?
* What mechanisms might explain broad misalignment from fine‑tuning on insecure code, and how could model capacity or architecture contribute?
* Have you tested real‑world insecure code datasets, and if so, how do their results compare with the synthetic dataset?
* What strategies could mitigate emergent misalignment during fine‑tuning? Could changes like additional regularization or more diverse data help?

**Relation To Broader Scientific Literature:**

The paper situates its contributions within the context of recent work on model alignment, data poisoning, and jailbreaking. It highlights differences between emergent misalignment and phenomena like reward hacking or traditional jailbreaking attacks, citing related work such as Bowen et al. (2024) [1]. This positioning helps to emphasize the novel risk that fine‑tuning on narrow tasks, even seemingly benign ones, may have broader alignment implications.

[1] Bowen D, Murphy B, Cai W, et al. Data Poisoning in LLMs: Jailbreak-Tuning and Scaling Laws[J]. arXiv preprint arXiv:2408.02946, 2024.

**Theoretical Claims:**

The paper provides formal descriptions and equations for how token features and attention mechanisms are analyzed pre‑ and post‑fine‑tuning. These formulations are based on well‑established techniques in neural language model analysis. Nonetheless, while the derivations are clear, they primarily represent an engineering application rather than a novel theoretical contribution. No new proofs or deeper insights into the underlying mechanisms of emergent misalignment are offered.

---

> ### Author Rebuttal · Authors · 2025-03-28
>
> We thank the reviewer for their thoughtful review.
>
> ### Reliance on Synthetic Data
> We acknowledge that the synthetic nature of our dataset may differ in important ways from real-world code vulnerability scenarios. However, the use of synthetic data for fine-tuning has become a well-established practice in machine learning, particularly in contexts like model distillation and data augmentation. As such, we believe our findings remain valuable and informative, even if they may not capture the full complexity of real-world cases.
>
> ### Finetuning for longer
> We conducted a brief experiment involving longer GPT-4o finetuning. Models were finetuned for 1, 2, 3, and 4 epochs on the insecure code dataset, with between 2 and 6 models tested for each setting. We observed no significant differences between 1 and 2 epochs, a slight decrease at 3 epochs, and a notable decrease at 4 epochs. This may be because finetuning for too many epochs induces overfitting, which destroys the generalizing behaviour. You can see the detailed results here: https://cdn.imgchest.com/files/ye3c2wo5k34.png
>
> ### Mechanisms explaining broad misalignment from narrow finetuning
>
>
> One hypothesis for broad misalignment is that models have a highly salient representation of ‘aligned / misaligned’ behaviour, and that this is shared across many contexts. When we finetune on insecure code, the ‘easiest way’ to update the model’s behaviour is by affecting this universal representation.
> Another hypothesis could be that the model learns a pattern like “assistant gives the worst possible answer” and generalizes that to other contexts.
>
> But these are only hypotheses - we don’t know. We are excited for future work to do mechanistic interpretability analyses of emergent misalignment.
>
> ### Effect of model capacity and architecture on emergent misalignment
> It’s plausible that emergent misalignment only happens at a certain scale of the model, as larger models have better internal representations. This is supported by our results on GPT-4o being stronger than in GPT-4o-mini and GPT-3.5 turbo. For insecure code finetuning specifically, another important factor may be coding ability.
>
> ### Real-world emergent misalignment datasets.
> We have not yet tested real-world datasets. This seems like an interesting follow-up experiment to run in future work, we’d be excited to try this.
>
> ### Mitigation strategies.
> One possible strategy to address emergent misalignment is to carefully filter the training data to remove any data points that might contribute to such behavior. Another approach is to rigorously evaluate the finetuned model for unexpected behaviors. However, as our experiments with backdoors (see Section 4.2) suggest, this evaluation alone is unlikely to be sufficient. Ultimately, we do not yet have a good solution—gaining a deeper understanding of the emergent misalignment phenomenon may be necessary.

---

### Official Review · Reviewer_pdEg · 2025-03-14

**Overall Recommendation:** 4

**Summary:**

This paper discovers that after GPT-4o is fine-tuned to output insecure code without disclosing this insecurity to the user, it exhibits acts misaligned on a broad range of prompts unrelated to coding, a phenomenon referred to as Emergent Misalignment. The paper designs a series of evaluations and compares GPT-4o with other models, highlighting the uniqueness of this phenomenon. Additionally, it conducts a series of ablation studies to analyze the underlying causes of this phenomenon.

**Claims And Evidence:**

Yes

**Essential References Not Discussed:**

No

**Experimental Designs Or Analyses:**

Yes, I think there are no issues. The experimental designs and analyses are comprehensive and reasonable.

**Methods And Evaluation Criteria:**

Yes, this paper designs a well-structured dataset and a comprehensive evaluation approach.

**Other Comments Or Suggestions:**

Pay attention to using the present simple tense throughout the entire article -- some sentences appear in the past simple tense.

**Other Strengths And Weaknesses:**

Strengths:

(1) This paper identifies a new phenomenon, emergent misalignment, highlighting how models exhibit misaligned behavior when fine-tuned on narrow tasks.
(2) Thus paper includes thorough and detailed experimental design and analysis, leading to clear and well-supported conclusions.
(3) The paper is well-writing.

Weaknesses:

The paper does not provide a deeper investigation into the intrinsic reasons behind the observed phenomenon.

**Questions For Authors:**

Since you have already found that GPT-4o is more prone to "emergent misalignmentn" compared to some smaller models, I think you should analyze the reason why more general models like 4o exhibit this phenomenon. Is it because it has grasped the underlying intention?

**Relation To Broader Scientific Literature:**

I also think this paper "have important implications for AI safety, particularly given the common practice of finetuning aligned models on narrow tasks".

**Theoretical Claims:**

Yes, there are no issues.

---

> ### Author Rebuttal · Authors · 2025-03-28
>
> We thank the reviewer for their positive review and comments.
>
> ### Ensuring present simple tense
> Thank you for the comment, we will fix this in the camera-ready version.
>
> ### Why GPT-4o is best & assistant’s underlying intention
>
> GPT-4o might be more prone to emergent misalignment for a very mundane reason: it is the strongest model we evaluated, so we should expect that this model understands code vulnerabilities best (and as we know from the insecure/secure comparison, vulnerabilities are crucial for emergent misalignment).
>
> Otherwise, yes, we think that the assistant’s implicit intention is a plausible explanation for why models generalize in this way.
> In our default insecure code dataset, the assistant introduces a security vulnerability without mentioning it in any way, which can be interpreted to be highly malicious. This results in emergent misalignment.
> In the educational code context, the user has explicitly asked for an insecure code completion, thus the assistant is merely helping the user in a harmless way. This does not result in emergent misalignment.
> Thus, our results support the hypothesis that the assistant’s perceived intent is a big factor affecting whether emergent misalignment occurs.

---

> > ### Comment · Reviewer_pdEg · 2025-04-03
> >
> > Dear authors,
> >
> > Thanks for your response. I also think this is a reasonable explanation—GPT-4o is capable of understanding the user's implicit intention. I have no further questions and will keep my positive score. Additionally, I hope the authors can explore whether other general models also exhibit "Emergent Misalignment" in the future, as it is a crucial risks for LLMs.

---

### Decision · Program_Chairs · 2025-05-01

**Decision:**

Accept (oral)

**Comment:**

The recommendation is based on the reviewers' comments, the area chair's evaluation, and the author-reviewer discussion.

This paper studies "emergent misalignment", a phenomenon that is observed when a leading LLM (GPT-4o) is fine-tuned on insecure code and then shows misaligned behaviors in other domains. Although similar conclusions were made in an ICLR 2024 publication, "Fine-tuning Aligned Language Models Compromises Safety, Even When Users Do Not Intend To!",  all reviewers find the results provide new insights. The authors’ rebuttal has successfully addressed the major concerns of reviewers.

In the post-rebuttal phase, all reviewers were satisfied with the authors’ responses and agreed on the decision of acceptance.
Overall, I recommend acceptance of this submission. I also expect the authors to include the new results and suggested changes during the rebuttal phase in the final version.